# Host Environment Shapes *S. aureus* Social Behavior as Revealed by Microscopy Pattern Formation and Dynamic Aggregation Analysis

**DOI:** 10.3390/microorganisms10030526

**Published:** 2022-02-28

**Authors:** Natsuko Rivera-Yoshida, Marta Bottagisio, Davide Attanasi, Paolo Savadori, Elena De Vecchi, Alessandro Bidossi, Alessio Franci

**Affiliations:** 1Department of Mathematics, Universidad Nacional Autónoma de México, Mexico City 04510, Mexico; n.riverayoshida@gmail.com; 2Laboratory of Clinical Chemistry and Microbiology, IRCCS Istituto Ortopedico Galeazzi, 20161 Milan, Italy; marta.bottagisio@grupposandonato.it (M.B.); attanasi.davide@libero.it (D.A.); elena.devecchi@grupposandonato.it (E.D.V.); 3Department of Endodontics, IRCCS Istituto Ortopedico Galeazzi, 20161 Milan, Italy; paolo.savadori@grupposandonato.it

**Keywords:** *S. aureus*, bacterial aggregates, biofilm, bacterial development, orthopedic infections, periprosthetic joints, joint infections

## Abstract

Understanding how bacteria adapt their social behavior to environmental changes is of crucial importance from both biological and clinical perspectives. *Staphylococcus aureus* is among the most common infecting agents in orthopedics, but its recalcitrance to the immune system and to antimicrobial treatments in the physiological microenvironment are still poorly understood. By means of optical and confocal microscopy, image pattern analysis, and mathematical modeling, we show that planktonic biofilm-like aggregates and sessile biofilm lifestyles are two co-existing and interacting phases of the same environmentally adaptive developmental process and that they exhibit substantial differences when *S. aureus* is grown in physiological fluids instead of common lab media. Physicochemical properties of the physiological microenvironment are proposed to be the key determinants of these differences. Besides providing a new tool for biofilm phenotypic analysis, our results suggest new insights into the social behavior of *S. aureus* in physiological conditions and highlight the inadequacy of commonly used lab media for both biological and clinical studies of bacterial development.

## 1. Introduction

Recent observations have questioned the dichotomous division of bacterial lifestyle in planktonic individuals and sessile biofilms on surfaces. Floating aggregates of bacterial cells have been described both in vitro and ex vivo from human chronic infection specimens [1,2,3,4,5]. Similarly to sessile biofilm, these planktonic “biofilm-like aggregates” show increased tolerance to antibiotics and are resistant to phagocytosis by immune cells. Understanding the interaction and the transition between the two bacterial lifestyles is therefore crucial to understanding the mechanisms of infections and treating them.

A common bacterium exhibiting both planktonic aggregate and sessile biofilm lifestyles is *Staphylococcus aureus*. *S. aureus* is able to adapt to different ecological niches and to provoke acute and chronic infections. The exceptional ability of *S. aureus* to adhere to abiotic surfaces positions it among the most common etiological agents retrieved from implanted medical devices [6]. Prosthetic joint infections (PJIs) are among the most challenging to treat, frequently requiring multiple antibiotic treatments and, in the case of failure, the removal of the prosthetic device, therefore significantly affecting patient morbidity and mortality rate [7].

A recent microscopic investigation from our group analyzing infected synovial fluids [1] confirmed *S. aureus* as the primary infection agent and corroborated in vitro findings describing its rapid aggregation in floating multicellular aggregates when inoculated in synovial fluids [8,9,10,11,12]. Evidence reported in the literature suggests that such phenotype is induced by host-environment-derived molecules, mainly polymers, and may not involve the expression of biofilm genes during its development [13,14,15]. Concomitantly, the appearance of planktonic aggregates has been associated with a marked phenotypic reorganization of the sessile biofilm forming on the surface exposed to the growing medium [1,8,16]. More precisely, when grown in physiological media, sessile biofilm appears as big sparse mounds of cells with a matrix composition closely resembling that of the above planktonic aggregates, exhibiting in particular a massive integration of medium-derived fibrin and hyaluronic acid [1,11,16,17]. Thus, chemical and physical properties of the microenvironment can shape bacterial social behavior and decrease susceptibility to antibiotics [1,10,12,18,19,20].

Here, we compared sessile biofilm and planktonic aggregate formation in a standard bacteriological culture medium (brain heart infusion—BHI) and in two biological fluids serving as physiological media (human plasma and bovine synovial fluid—BSF), under different oxygen availability conditions, and in the presence or absence of motion stress. Using optic and confocal microscopy, image analysis tailored to quantify fine patterning differences, and mathematical modeling, we explored the role of physicochemical properties of growth media in shaping planktonic aggregation kinetics, sessile biofilm patterning, and the transition from the former to the latter. We also analyzed how oxygen availability and the presence of motion stress affect these developmental processes. Finally, we built and analyzed a minimal computational model of staphylococcal planktonic aggregation and sessile biofilm patterning. Our model solely includes generic physical forces whose strength depends on the simulated medium physicochemical properties. Yet, it is sufficient to qualitatively reproduce and mechanistically explain medium-dependent planktonic aggregation kinetics and sessile biofilm formation patterns.

## 2. Results

### 2.1. Physicochemical Medium Properties Affect Biofilm Surface Distribution

Commercial broths, such as BHI, are the most commonly used media for in vitro growth of *S. aureus* biofilm, usually in static and aerobic conditions. To more trustfully study *S. aureus* biofilm formation in PJIs, we used a mixture of ex vivo and in vitro experimental designs, in which bacteria collected ex vivo from infected prosthetic knees were grown either in a conventional culture medium (BHI) or in physiological media (plasma and BSF) under different oxygen availability (aerophily, microaerophily) and in the presence or absence of motion stress (static, agitation). See Section 4 for details.

The sessile biofilm grown in each setting was investigated after 48 h through confocal laser scanning microscopy (CLSM—see Section 4). Already at first glance, CLSM images evidenced that the medium alone was sufficient to drastically modify *S. aureus* sessile biofilm phenotype (Figure 1A,B). Laboratory medium led to a dense and homogeneous biofilm distribution, while the surface occupied by biofilm was less and more irregularly distributed in physiological media, with relatively larger biofilm mounds in BSF, and relatively smaller mounds in plasma.

We quantified these observations in Figure 2. In all oxygen availability and motion stress conditions, the total surface occupied by sessile biofilm was significantly larger when comparing BHI with physiological media (Figure 2A, Appendix A). Conversely, there were no significant differences when comparing plasma and BSF. No significant differences were observed when comparing biofilm surface in the same medium but in different oxygen availability and motion stress conditions (Figure 2A, Appendix A). On average, BHI led to slightly larger biofilm volume (Figure 2B). Statistical analysis was not performed for volume comparisons. We finally compared the ratio between biofilm volume density (i.e., μm^3^ of biofilm volume per μm^2^ of titanium surface) and biofilm surface density (i.e., μm^2^ of biofilm surface per μm^2^ of titanium surface). In all oxygen availability and motion stress conditions, BSF led to a higher ratio than BHI and plasma (Figure 2C). This result suggests robust, condition-independent differences in the spatial pattern of biofilm distribution—more packed and concentrated in BSF compared with BHI and plasma. We will assess this last observation quantitatively through pattern formation analysis in the next section.

### 2.2. Physiological Media Favor a Heterogeneous Biofilm Pattern Formation

To quantify possible differences in the spatial pattern of biofilm distribution, we segmented each CLSM image into grids of 15×15 elements (Figure 3A) and calculated the local biofilm surface density in each of the resulting grid elements (Figure 3B and Appendix A). We then compared the resulting distributions of biofilm densities across the grid elements (Figure 3C,D and Appendix A) as a function of the used medium, oxygen availability, and motion stress conditions. Consistent with the representative CLSM images shown in Figure 3A, both BHI and plasma led to a roughly unimodal probability density functions (PDFs). In BHI, the distribution was basically unskewed, as the small difference between the mean and median revealed. Relative variations of biofilm density were larger in plasma than BHI, as reflected by wider PDFs. Conversely, BSF led to bimodal and skewed biofilm density PDFs, charged more heavily toward low biofilm densities as compared with both BHI and plasma.

These observations were independent of the number of grid elements used to compute density distribution (Appendix A). Statistical analysis of biofilm density distributions revealed a marked sensitivity of bacterial development to changes in the used medium in almost all the oxygen availability and motion stress conditions (Appendix A, Figure 3D). Comparing the biofilm density PDFs for different media and the same oxygen availability and motion stress condition led to significant differences in all cases except two (BHI vs. plasma in aerobic agitated condition, BSF vs. plasma in microaerobic agitated condition). Statistical analysis also revealed that the sensitivity of bacterial biofilm development to oxygen availability and motion stress conditions depend on the medium (Appendix A, Figure 3C). Sessile biofilm-forming cells cultured in BHI were basically insensitive to changes in oxygen availability and motion stress conditions, as reflected by the lack of significance for all the comparisons performed between biofilm density distributions. Conversely, for BSF, biofilm density distributions were highly sensitive to oxygen availability and motion stress conditions, as reflected by significant differences for all the comparisons. Finally, sensitivity of sessile biofilm grown in plasma led somewhat between the BHI and BSF cases, with only half of the conditions pairs (microaerobic static vs. microaerobic agitated, aerobic agitated vs. microaerobic agitated) being distinguishable by statistical significance. Inspection of mean and standard deviation of biofilm density (Appendix A) further confirmed increased sensitivity to both uncontrolled (across replicas) and controlled (across oxygen availability and motion stress conditions) parameter variations as the medium changed from BHI to plasma to BSF.

Pattern formation was also quantified using spectral methods by analyzing the spectral density functions (SDFs) of CLSM images’ two-dimensional fast Fourier transform (Figure 4A,B). The analysis is favorably performed by comparing the resulting cumulative spectral power functions (CSPfs), computed by integrating SDFs over increasingly larger squares in the frequency domain (Appendix A). Integration in the frequency domain makes CSPfs analysis less sensitive to noise as compared with SDFs analysis. A CSPf close to the diagonal of the rectangle $\left[0,\;\lambda_{max}\right]\times \left[0,\;1\right]$, where $\lambda_{max}=\frac{512}{2.325}\;{\mathrm{mm}}^{-1}$ is the maximum spatial frequency of the discrete Fourier spectrum (images capture $2.325\times 2.325\;{\mathrm{mm}}^{2}$ of surface and where captured at a resolution of 1024×1024 pixels), which corresponds to a spatial pattern with all spatial frequencies roughly equally represented, that is, closely resembling a spatial white noise. Departure of a CSPf from the diagonal reflects a more structured spatial pattern. More precisely, a downward concave CSPf reflects a higher power in low spatial frequencies, corresponding to a more patchy spatial distribution. For all oxygen availability and motion stress conditions, and for all replicas, BHI robustly led to roughly diagonal CSPfs (Figure 4C), reflecting the close resemblance between biofilm distribution in BHI and spatial white noise. Moving on to plasma and BSF, CSPfs become increasingly downward concave. Furthermore, in physiological media, CSPfs exhibit increasingly large variability across oxygen availability and motion stress conditions, and across replicas. Statistical analysis of CSPfs confirmed the sensitivity of bacterial biofilm development to environmental conditions. CSPfs were significantly different when comparing growing media in the same oxygen availability and motion stress conditions in all but three cases (plasma vs. BSF in aerobic agitated condition; plasma vs. BSF in microaerobic agitated conditions; plasma vs. BHI in microaerobic static conditions) (Figure 4D, Appendix A). When comparing the same growing medium in different oxygen availability and motion stress conditions, CSPf analysis reaffirmed increased sensitivity in physiological media (half of conditions pairs led to significantly different CSPf) as compared with lab medium (no significant comparisons) (Figure 4C, Appendix A).

The contribution of this section was twofold. On the one hand, the results in this section provided biological evidence that the growth medium has profound effects on sessile biofilm spatial distribution and on its sensitivity to other environmental conditions (oxygen availability, motion stress). In particular, lab media led to more random and more insensitive biofilm distributions as compared with physiological media. On the other hand, the methodology used to obtain these results (local and spectral statistical analysis of CLSM images) constitutes, to the best of our knowledge, a novel approach in the context of bacterial development.

### 2.3. Physiological Media Induce Planktonic Aggregation and Shapes Transition to Sessile Biofilm

The kinetics of planktonic aggregation revealed further profound differences between growing media (Figure 5A). When *S. aureus* was grown in BHI, cells exhibited what is considered a standard planktonic growth, characterized by floating aggregates, seldom exceeding the area of 3–4 $\mu m^{2}$, which is the average projected size of a staphylococcal tetrad (Figure 5B). A sharp increase in the number of these small aggregates was observed between 4 and 6 h of incubation, reflecting a late exponential phase that started at around 3 h (Appendix A). Plasma led to the rapid appearance of large aggregates already after 4 h, whose size kept on increasing until 6 h of incubation and then plateaued. The total number of aggregates in plasma was non-monotonic in time. It first decreased due to early aggregation, but it started to rapidly increase between 6 h and 8 h of incubation after entering a late exponential phase at around 4 h (Appendix A). In BSF, there was a clear inverse relationship between the aggregate average size and the total number of aggregates, similarly to that observed in plasma until 6 h of incubation. At 8 h, the effects of the late exponential phase (which started at 6 h in BSF—Appendix A) were still not appreciable.

The kinetics of sessile biofilm formation happening beneath planktonic aggregation also revealed profound differences between media (Figure 5B), whereas in BHI an extremely large number of small sessile biofilm mounds could be appreciated already after 8 h, in plasma 24 h, and in BSF up to 48 h passed before the appearance of an appreciable amounts of sessile biofilm. The difference between lab and physiological media in the kinetics of sessile biofilm formation likely reflects the sharply different planktonic aggregation kinetics in BHI as compared with physiological media. The rapid saturation of lab growing medium, due to its earlier and stronger exponential phase, driven by the superabundance of nutrient as compared with physiological media, together with the rapid precipitation of bacteria toward the titanium plate due to very low viscosity of lab medium, were likely the key determinants of the extremely fast appearance of abundant, but small, sessile biofilm mounds in BHI. The further large increase in BHI sessile biofilm observed between 24 h and 48 h might be due to the depletion of nutrients in the growing medium and the subsequent transition to a phenotype more prone to surface attachment. When comparing the two physiological media, the two key differences were in the temporal course of sessile biofilm development, earlier in plasma as compared with BSF, and in the average size of biofilm mounds, smaller and not increasing in time in plasma, larger and increasing in time in BSF. Furthermore, whereas in plasma sessile biofilm mounds remained about 10 times smaller than the above planktonic aggregates, in BSF, they reached roughly half the size of the above planktonic aggregates at 48 h.

### 2.4. Medium Viscosity and Bridging Forces Shape Planktonic Aggregation and Its Transition to Sessile State

The behavior of the computational model Equation (Equation 1) is determined by only three-dimensional parameters, $k_{int}$, *D*, $v_{g}$, respectively, associated with three physicochemical properties of the medium: the strength of bridging forces between bacteria, the diffusion coefficient, and the gravitational drift toward the titanium plate (Figure 6A). The three parameters are jointly determined by the concentration of polysaccharides, proteins, and other physiological molecules in the medium, while the presence and strength of bridging forces mainly depends on the presence and abundance of fibrinogen and albumin, and diffusion coefficient and gravitational drift are determined by the medium viscosity, which is mainly altered by the presence and abundance of polysaccharides and proteoglycans (see Table 1 for more details and references). It follows that in BHI bridging forces should be close to zero, while both diffusion coefficient and gravitational drift should be relatively large. Conversely, physiological media should be characterized by the presence of substantial bridging forces and relatively small diffusion coefficient and gravitational drift. Using the documented difference in their molecular compositions (Table 1) and bridging forces, diffusion and gravitation drift should be larger in plasma as compared with synovial fluid. The resulting physicochemical properties of the three media are schematized in Figure 6A.

In simulations, we use $k_{int}=0.1,v_{g}=0.2,D=0.1$ for BHI, $k_{int}=50.0,v_{g}=0.05,D=0.05$ for plasma, and $k_{int}=25.0,v_{g}=0.02,D=0.02$ for BSF. Appendix A reproduce the model behavior for BHI, plasma and BSF parameter regime, respectively. In order to keep the model computationally manageable, we only simulated the evolution of a thin horizontal slice of the whole bacterial community. Observe that the three forces at play have complementary effects on the collective bacterial behavior. Diffusion favors disgregation but also random getting together, while bridging forces favors cells staying together. The role of gravitational drift is mostly to affect the time spent in planktonic state before reaching the titanium plate—that is, the time during which diffusion and bridging forces can effectively affect bacterial behavior. In BHI, bacteria rapidly fall toward the titanium plate in a roughly random pattern. In plasma and BSF, the time spent in planktonic state is much larger, which permits the emergence of stable planktonic aggregates. In plasma, stronger diffusion and bridging forces, as compared with BSF, favor the earlier appearance of bacterial aggregates (Appendix A). However, the sticking effect of bridging forces is also rapidly balanced by the disgregating effect of diffusion in plasma. Conversely, in BSF, bacterial aggregates take more time to appear due to weaker diffusion and bridging forces. However, once formed, slower gravitation drift and lower diffusion allow the planktonic aggregates to grow further before reaching the plate. Overall, the time course of the collective bacterial planktonic behavior in our model qualitatively captures and provide a mechanistic explanation for the kinetic analysis of planktonic aggregation reported in Figure 5.

The ability of our model to explain our experimental observations can further be assessed by comparing the actual and simulated sessile biofilm distribution (Figure 6B). We carried this out by comparing how simulated and experimentally measured biofilm density CSPfs change as a function of the actual or simulated growing medium (Figure 6C). At first glance, as shown in Figure 6C, in both model and experiments, biofilm density CSPfs qualitatively change from robustly diagonal—i.e., spatial white noise-like—for BHI, to increasingly downward concave, and with increased sensitivity—i.e., increased variability—to random parameter variations across replicas, for physiological media. Quantitatively, statistical comparison of biofilm CSPfs obtained from different simulated media led to the same significant differences as for experimentally measured CSPfs (Appendix A).

The behavior of our simple model, solely governed by viscosity and bridging forces-related parameters, matches qualitatively a large part of the experimental observations we made on staphylococcal planktonic aggregation, sessile biofilm formation, and the transition from the former to the latter. The quantitative discrepancies between our model and experiments could probably be overcome with more detailed (e.g., agent-based) models. However, simple qualitative modeling provides strong evidence that the interaction between medium viscosity and bridging forces is a key determinant of planktonic aggregation and of the transition from planktonic to sessile state.

## 3. Discussion

Because of their subtle symptomatology and treatment recalcitrance, PJIs are true clinical challenges. Biofilm formation is thought to be a key player of this type of bacterial recalcitrance [10,12,13,27]. Our previous investigation, based on ex vivo analysis [1], suggested that staphylococcal behavior is sharply different in physiological fluids as compared with commercial broths, particularly in two aspects: the presence of planktonic aggregates and a considerable phenotypic reorganization of the sessile biofilm. This evidence raised the need to further investigate the dynamics of planktonic aggregation and sessile biofilm formation in complex biological fluids, in order to understand the origin of PJIs and how to treat antibiotic-resistant infections.

We contributed to this task by applying novel CLSM image analysis methods to characterize in vitro bacterial development and by deriving a mathematical model to validate our results. The novelty of our CLSM image analysis method relies in that it is tailored to quantify patterning and not only globally averaged measures. For this study, we employed a clinical strain isolated from an infected joint that was characterized in a recent investigation [1]. The lack of genetic characterization and the employment of a single clinical isolate represent a known limitation. Thus, the complex social behavior of *S. aureus* we observed needs to be extended to other staphylococcal strains, including genetically controlled ones.

First, using three-dimensional reconstruction and two-dimensional projections of CLSM images, we revealed that the commercial lab medium BHI supported a much thinner and widely spread sessile biofilm growth when compared with physiological media such as plasma and BSF, confirming and extending previous similar observations [1,8,16]. Particularly in BSF, biofilm appeared in sparse but thick patches, known to confer antimicrobial resistance by protecting inner cells [28]. This provides a mechanism for the increased biofilm resistance to antimicrobial agents in the presence of BSF. We also discovered that the sensitivity of sessile biofilm phenotype to physiologically relevant environmental conditions, such as oxygen availability and motion stress also depends on the growth medium. Null sensitivity was observed in the lab medium BHI. This observation reasserts the masking effect of lab media on bacterial phenotypic variation previously described in the literature [29,30]. Conversely, biofilm formation was highly sensitive to environmental conditions in BSF, while plasma led to an intermediate sensitivity.

Second, we quantified the kinetics of planktonic biofilm aggregation and of the concomitant sessile biofilm formation. Consistent with previous reports [8,31], we showed that planktonic aggregation in plasma and BSF is very fast and involves half of the bacterial population already after 4 h of inoculum. Conversely, planktonic aggregates could still not be observed in lab medium BHI after 8 h of inoculum. A possible explanation for this observation is the absence of host-derived bridging molecules (i.e., fibrinogen). In their absence, cells must produce intercellular adhesion factors, whose expression is probably conditioned to extreme environmental stresses, such as starvation [32,33]. On the other hand, staphylococci express an impressive arsenal of surface adhesins, which allows them to scavenge host-derived polymeric molecules and to exploit them as an extracellular matrix [1,8,11,13,21]. Moreover, other mechanisms are known to be involved in bacterial aggregation and biofilm formation, such as the homophilic interactions of surface-expressed proteins [34,35]. The kinetics of sessile biofilm formation happening beneath planktonic aggregation suggests a delayed “fall-and-stick” mechanism in physiological media. According to this mechanism, small planktonic aggregates would rapidly reach and stick to the surface, while larger aggregates would take longer to reach the surface and attach to it due, for instance, to the “cellular crowd” hindrance in the medium or to its viscosity. This mechanism is partially confirmed by the observation that sessile biofilm grown in BSF presented a similar extracellular matrix phenotype to planktonic aggregates formed above it [1,13]. In other words, we propose that in physiological media, planktonic and sessile multicellular communities are not developed independently but are rather two interlinked phases of the same developmental process. This two-phase developmental process brings some clear evolutionary advantages: a rapid planktonic aggregation provides a first shelter from immune cells, which are unable to phagocytize aggregates of cells exceeding their size [5]. This also gives the bacterial community time for the subsequent colonization of the implanted biomaterial and the progression to chronic infection through the formation of sessile biofilm.

Third, to test whether purely generic physical medium properties are sufficient to reproduce our results, we developed a minimal mathematical model. Our model is solely parameterized by the modeled medium viscosity and abundance of bridging molecules, i.e., favoring cell-to-cell interaction [14]. Of the three growing media used in this work, BHI neither contains bridging molecules nor is viscous, while plasma and BSF are both rich in bridging molecules and viscous—although to different degrees. Plasma is richer in bridging molecules, particularly fibrinogen, fibronectin, and albumin [21], but has lower viscosity. BSF is more viscous, particularly due to the abundance of hyaluronic acid (HA) [23], absent in plasma. Implementing these observations in our mathematical model, we were able to reproduce all the qualitative differences observed in planktonic aggregation and sessile biofilm formation in the three different media. The fundamental role of generic physical processes for bacterial development is well known [36,37]. Specifically, the active mechanical interaction of the microorganisms with their environment has been shown to alter their spatial organization from the single cell scale to the population scale [19,20]. Here we confirmed these key findings also for infectious staphylococcal development.

Our work has both biological and clinical implications, the main one of which is that lab media can lead to both biologically and clinically biased conclusions. We propose that planktonic aggregation is not an alternative state to sessile biofilm. In the biologically relevant environmental niches where *S. aureus* is found, planktonic aggregation is rather a precursor and determinant of the sessile biofilm formation. This observation would have been hard or impossible to make by growing bacteria in lab media. This is a direct consequence of the presence of host-derived molecules in the physiological media and their rapid exploitation as a matrix by staphylococci. From a clinical perspective, these facts have clear implications for the relevance of commonly used therapeutic strategies, usually developed and tested in lab conditions. As an example, poly-N-acetylglucosamine (PNAG) is commonly found as an extracellular matrix component of staphylococcal biofilm-like aggregates in synovial fluid through staining with WGA lectin [8,10]. However, WGA is a lectin known to bind to GlcNAc residues that are present both in HA and in PNAG, thus it is impossible to distinguish the two polymers in the aggregates matrix. Indeed, HA incorporation in biofilm-like aggregates was directly demonstrated by staining with WGA lectin aggregates of staphylococcal cells lacking the *ica* operon [11] and it has been reported that PNAG production is redundant in staphylococci growing in plasma [15]. This would imply that what has been considered the main staphylococcal biofilm matrix component may not be present in clinical biofilms and that, in their stead, host-derived polymers would be the key players to look after, as crucially pointed out by our analysis. Thus, any efforts made to target biofilm grown in commercial lab media by acting enzymatically on the matrix might be vain.

This work solely focused on developing and fine-tuning a novel model for microscopy images analysis to describe bacterial phenotypical behavior. Additional studies comprising well-characterized, genotypically different lineages are needed to shed further light on the mechanisms of bacterial adaptation to changing environment and their biological and clinical implications and to validate the preliminary speculations made thank to the model proposed in the present work.

## 4. Materials and Methods

### 4.1. Bacterial Strain and Culture Conditions

A clinically relevant strain isolated from an infected prosthetic knee at the Laboratory of Clinical Chemistry and Microbiology of the IRCCS Galeazzi Orthopedic Institute was used. In particular, a multi-drug-resistant high biofilm producer *S. aureus* strain was chosen. Identification and antimicrobial susceptibility testing was carried out on a Vitek2 Compact (BioMerieux) and confirmed by sequencing of 80 bp of the variable regions V1 and V3 of the 16S rRNA gene by Pyrosequencing (PSQ96RA, Diatech). The strain was stored at −80 °C in BHI broth (Merck) enriched with 10% glycerol (VWR Chemicals) until used. Planktonic and sessile growth was performed throughout the work in BHI broth (Merck), human plasma, and bovine synovial fluid (BSF, Lampire Biological Laboratories). Human plasma was obtained from healthy donors, pooled, and stored at −20 °C until use. Plasma samples were collected exclusively from some of the authors of the present work, no patients were enrolled for this research purpose. Hence, no informed consent from the participants or ethics committee approval was necessary to authorize the use of this voluntary contribution in this specific case. Bacterial cells were reconstituted from frozen stocks by seeding on Tryptic Soy Agar (TSA, Biomérieux) and incubating the plates overnight at 37 °C. Then, the inoculum was prepared by resuspending single colonies in sterile saline solution at turbidity of 0.5 McF (approximately 1.5 $\times \;10^{8}$ CFU/mL). The 5 mm diameter titanium alloy disks (Ti6Al4V, Geass) were placed in a flat-bottom 96-well microplate (VWR) and covered with 180 mL of either sterile BHI, BSF, or plasma. Then, 20 µL of the inoculum were dispensed in each well to reach a final bacterial concentration of ∼10${}^{7}$ CFU/mL. To evaluate the contribution to phenotypic changes by physical conditions that mimic the physiological environment, the microplates were incubated at 37 °C either in aerobiosis or in microaerophily, by placing the plate in a in a microaerophilic chamber (AnaeroPack-MicroAero Gas Generator, Thermo Fisher Scientific, Rodano, Italy) [12], and either in static or dynamic conditions, by placing the plate on a shaker at 200 rpm.

### 4.2. Confocal Microscopy of Sessile Biofilms

After appropriate time of incubation, titanium disks were gently washed 3 times with sterile saline and stained with Syto^™^ 9 green-fluorescent nucleic acid stain (Thermo Fisher Diagnostics SpA). Briefly, the staining solution was prepared by adding 1 µL of Syto^™^ 9 to 1 mL of filter-sterilized water and 30 µL were applied all over the surface for 30 min in the dark, then washed again, and allowed to dry until analysis. The disks were directly examined with an upright TCS SP8 (Leica Microsystems CMS GmbH) using a 20× objective (HC PL FLUOTAR 20×/0.50) or a 5× objective (HC PL FLUOTAR 5×/0.15). A 488-nm laser line was used to excite Syto^™^ 9. Images from at least three randomly selected areas were acquired for each surface of four independent replicas. The obtained images were processed with LAS X software (Leica Microsystems CMS GmbH) and analyzed by Fiji software (Fiji, ImageJ; Wayne Rasband National Institutes of Health).

### 4.3. Analysis of Planktonic Bacterial Cells

To analyze planktonic cells, the strain was inoculated as described above and aliquots were withdrawn after 2, 4, 6, and 8 h using a sterile 10 µL loop and a drop was placed on a microscope glass slide (SuperFrost, Thermo Scientific, Rodano, Italy). The drop was then fixed by covering with paraformaldehyde 4% (Carlo Erba) and dried on a hot plate at 50 °C. Cells were then stained with crystal violet (Carlo Erba) and analyzed by an upright optical microscope (Olympus CX43) with a 40× objective. For each time point, three drops were withdrawn from each tube. The experiment was performed in triplicate for each strain. Five random images were acquired from each sample.

### 4.4. Growth Curves

To obtain the growth curves in BHI, human plasma, and BSF, cells were resuspended in sterile saline from an overnight growth in TSA plates and inoculated in the wells of a 96-well microplate to reach a concentration of ∼10${}^{6}$ CFU/mL. Optical density was read at 595 nm wavelength every 30 min by means of a microplate reader spectrophotometer (Multiskan FC, Thermo Fisher Scientific). The content of each well was thoroughly resuspended by energetically pipetting to disperse any forming aggregates. Four replicates were performed for each media.

### 4.5. Computational Model

To model the effect of the medium physicochemical properties on bacterial aggregation, a group of *N* bacteria was considered in a two-dimensional liquid space above a titanium plate: the first dimension describes the vertical position of the bacteria and the second dimension describes their horizontal position. $z_{i}\left(t\right)$ represents the height of bacterium, *i*, above the titanium plate at time *t*. $x_{i}\left(t\right)$ represents the horizontal position of bacterium *i* at time *t*. Let $X_{i}=\left(x_{i},z_{i}\right)$. In a strong-friction limit, the equations of motion for each bacterium sufficiently far from the titanium plate (i.e., for $z_{i}>0$ and sufficiently large) are as follows:(1)$$\begin{array}{c} \begin{array}{cc} dx_{i} & =\left(k_{int}\sum_{\begin{array}{c} j=1\\ j\neq i \end{array}}^{N}I_{x}\left(X_{i}-X_{j}\right)\right)dt+DdW_{xi},\;i=1,\ldots ,N,\\ dz_{i} & =\left(k_{int}\sum_{\begin{array}{c} j=1\\ j\neq i \end{array}}^{N}I_{z}\left(X_{i}-X_{j}\right)-v_{g}\right)dt+DdW_{zi},\;i=1,\ldots ,N, \end{array} \end{array}$$
where $k_{int}$ is the bacterial interaction strength due to bridging forces, the function $I\left(X_{i}-X_{j}\right)=\left(I_{x}\left(X_{i}-X_{j}\right),I_{z}\left(X_{i}-X_{j}\right)\right)$ models the short-range bridging force exerted by bacterium *j* over bacterium *i*, $v_{g}$ is a vertical drift due to gravity, *D* is the diffusion coefficient, and $dW_{ix},dW_{iz}$ are independent Wiener processes. For the short-range interaction function we use Equation (Equation 2).
(2)$$I\left(X_{i}-X_{j}\right)=\left\{\begin{array}{cc} -\min\left(e^{-\frac{∥X_{i}-X_{j}∥}{b_{size}}}e^{\frac{1}{∥X_{i}-X_{j}∥}},10\right)\frac{\left(X_{i}-X_{j}\right)}{∥X_{i}-X_{j}∥}, & \mathrm{if}\;∥X_{i}-X_{j}∥\geq b_{size}\\ 100\frac{X_{i}-X_{j}}{\max\left(∥X_{i}-X_{j}∥,\frac{b_{size}}{100}\right)}, & \mathrm{if}\;∥X_{i}-X_{j}∥<b_{size} \end{array}\right.$$

The parameter $b_{size}$ models bacterium size. The short-range interaction force is attractive when the distance between two bacteria is larger than $b_{size}$ and becomes strongly repulsive when two bacteria are closer than $b_{size}$, which avoids cell overlapping. Because of the first decreasing exponential in the expression of $I(X_{i}-X_{j})$ for $∥X_{i}-X_{j}∥\geq b_{size}$, the interaction force is short range. In particular, it becomes exponentially small with characteristic spatial scale $b_{size}$ as the distance between the two bacteria increases. The second exponential lets the interaction strength increase exponentially when the two bacteria are close enough. For numerical stability, we bound the interaction strength. The chosen upper limit in simulations is 10(a.u.). Similarly, we let the repulsive force strength at short distances be equal to 100. The max function at the denominator for $∥X_{i}-X_{j}∥<b_{size}$ is again used for numerical stability. Note that because we are interested solely in modeling the effect of the medium physicochemical properties on bacterial aggregation and to maintain the model minimal, we do not explicitly include cellular growth in our model. We also assume that all the forces rapidly decreases to zero as the bacterium approaches the titanium plate, that is, we assume that in the sessile state bacterial movement is negligible as compared to the planktonic state. We achieve this by multiplying the vector field components by $\tanh(z_{i})$ and by imposing ${\dot{z}}_{i}=0$ for $z_{i}<10^{-3}$.

For numerical simulations, we used N=150, $b_{size}=0.1$, initial conditions $x_{i}\left(0\right)\in U\left(-2.5,2.5\right)$, $z_{i}\in N\left(5.0,0.1\right)$, where U(a,b) denotes the uniform distribution between *a* and *b* and N(a,b) denotes the normal distribution with average, *a*, variance, *b*, and parameters, as specified in the text. This choice for initial conditions was meant to simulate one thin horizontal slice of the whole bacterial community. All simulations were performed with the Julia DifferentialEquations SDE solver with Euler–Maruyama method and time-step dt=0.1. Biofilm density in Figure 6 was computed by counting the number of simulated bacteria over sliding horizontal windows of length 0.1 and sliding step of 0.08. The simulated biofilm surface was computed by counting all space points where biofilm density was larger than 0.01. CSPfs associated with biofilm density were computed by integrating biofilm spectral power obtained through the Julia FFT funtion. For comparison with experimental data, the simulated CSPf arrays were interpolated to 512 points, i.e., the same dimension of experimental CSPf arrays. Biofilm density heatmaps in Appendix A were computed by counting the number of simulated over sliding rectangular windows of dimensions 0.2×0.5, horizontal sliding step 0.08, and vertical sliding step of 0.22.

To determine how diffusion coefficient and gravitation drift varies as a function of the simulated media, let η be the medium dynamic viscosity, ρ its density, $\mu =\frac{\eta }{\rho }$ its kinetic viscosity, and [w] the concentration of a (poly)carbohydrate, such as hyaluronic acid, or other viscosity-increasing molecule. Experimental results on the viscosity of aqueous carbohydrates solutions [38,39] suggest that $\mu \propto e^{\left[w\right]}$. Considering that dissolving carbohydrates in water linearly increases the mass without altering the solution volume, we have $\rho =\rho_{0}+c_{\rho }\left[w\right]$, which we approximate for simplicity as $\rho =\rho_{0}$ (i.e., we neglect dependence of medium density on [w]). By Stokes–Einstein equation $D\propto \eta^{-1}$ and by definition of kinematic viscosity $v_{g}\propto \mu^{-1}$. Thus, both $D\propto e^{-[w]}$ and $v_{g}\propto e^{-[w]}$, that is both diffusion and gravitation drift become exponentially small as the concentration of hyaluronic acid and other polymers, saccharides, and proteoglycans increases.

### 4.6. Statistical Analysis

To test whether the media influenced biofilm surface and volume a Wilcoxon test was performed (p<0.05, (Figure 2A,B; Appendix A). To test if oxygen availability and motion stress contributed to the surface biofilm attachment and its volume a Wilcoxon test was performed (p<0.05, (Figure 2A,B; Appendix A). For the statistical comparison of the probability density function curves in the pattern analysis (Figure 3C,D and Appendix A), Wilcoxon test (p<0.05) was conduced. For the Fourier analysis in Figure 4, each CLSM image was converted to a 8-bit grayscale image and the Fourier spectrum was obtained subsequently by the fast Fourier transform function (Figure 4B). The cumulative spectral power was calculated by adding the Fourier spectrum matrix elements concentrically (Figure 4C and Appendix A). Kolmogorov–Smirnov test was employed for the statistical comparison across the cumulative spectral power functions (p<0.05; Figure 4C,D; Appendix A). To compare between the experimental and the modeled cumulative spectral power functions, the Kolmogorov–Smirnov test was performed (p<0.05; Figure 6C; Appendix A). All analyses were conducted in R program (v. 3.2.3) through RStudio. ggplot2 package v. 3.0.0 was employed for visualization.

## Figures and Tables

**Figure 1 microorganisms-10-00526-f001:**
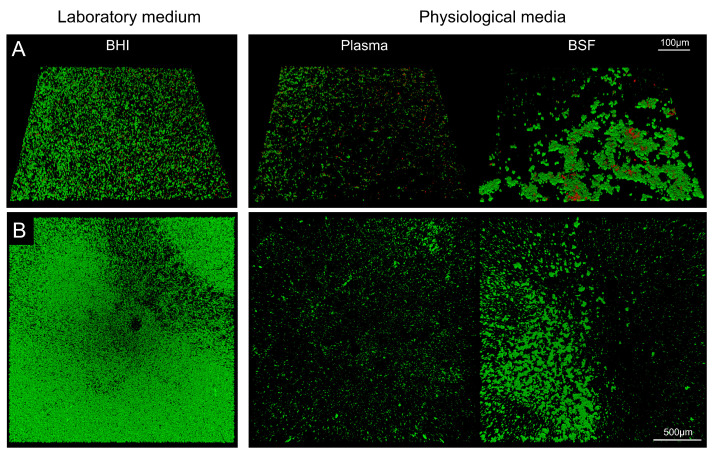
Three-dimensional reconstruction of sessile biofilm formation on titanium surface by means of CLSM. (**A**) Inclined view and (**B**) top view of biofilm grown in the presence of laboratory medium BHI or in physiological fluids, plasma (**left**) and bovine synovial fluid (BSF, **right**). **Upper** panels show images acquired with a 20× objective; **lower** panels show images acquired with a 5× objective.

**Figure 2 microorganisms-10-00526-f002:**
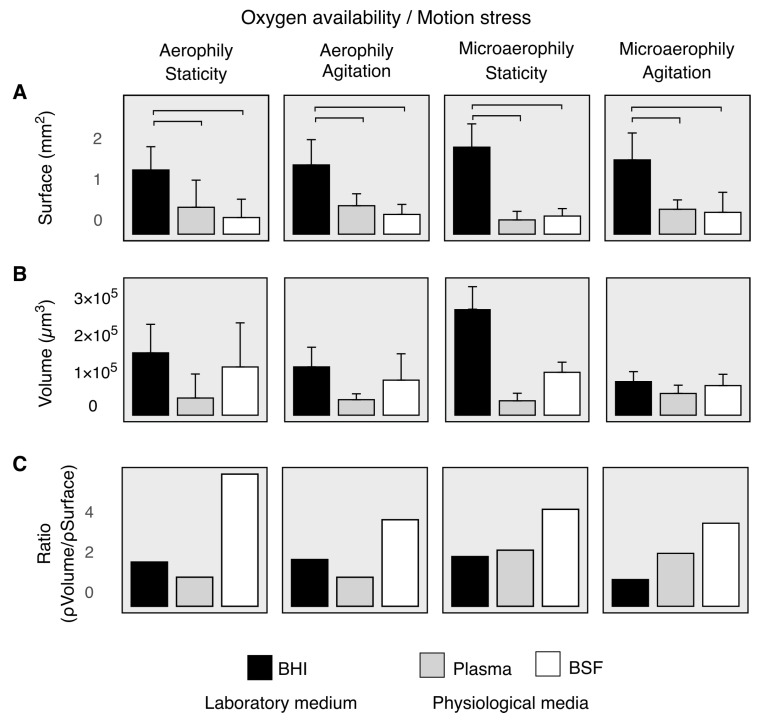
CLSM quantitative analysis of biofilm volume and surface on titanium disks demonstrates a reorganization of the sessile architecture and biomass distribution. (**A**) Total surface occupied by sessile biofilm on titanium disk. Horizontal lines represent statistical significance (Wilcoxon, p<0.05). (**B**) Total volume of biofilm biomass. (**C**) Average volume/surface ratio in the different media. Black bars represent biofilm grown in laboratory medium BHI, gray and white bars represent biofilm grown in physiological fluids, respectively, plasma and BSF.

**Figure 3 microorganisms-10-00526-f003:**
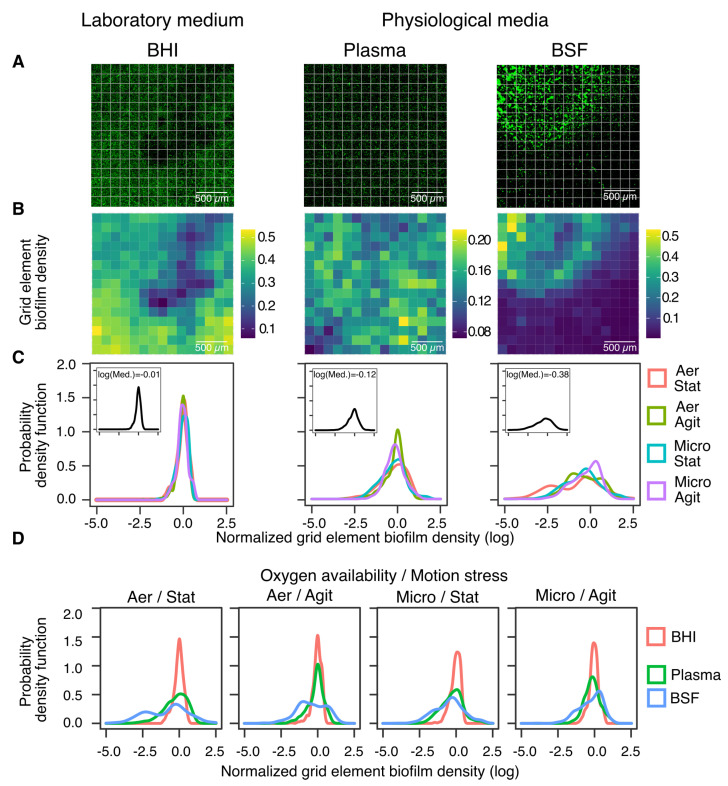
Pattern formation analysis reveals significant differences between lab and physiological media. (**A**) Representative CLSM images of sessile biofilm attachment to a titanium disk submerged in laboratory medium (BHI) or in physiological media (plasma, BSF). The 15×15 grid used for analysis is also shown. (**B**) Heatmap showing the biofilm density in each grid element. (**C**) Probability density function of grid element biofilm density for the same medium and different oxygen availability and motion stress conditions. Grid element biofilm density was normalized with respect to the mean biofilm density of each image to highlight changes in biofilm distribution patterns independently of mean biofilm growth. The insets show PDFs averaged across oxygen availability and motion stress conditions and report the median of the distribution. (**D**) Probability density function of grid elements biofilm density for different media and same oxygen availability and motion stress conditions.

**Figure 4 microorganisms-10-00526-f004:**
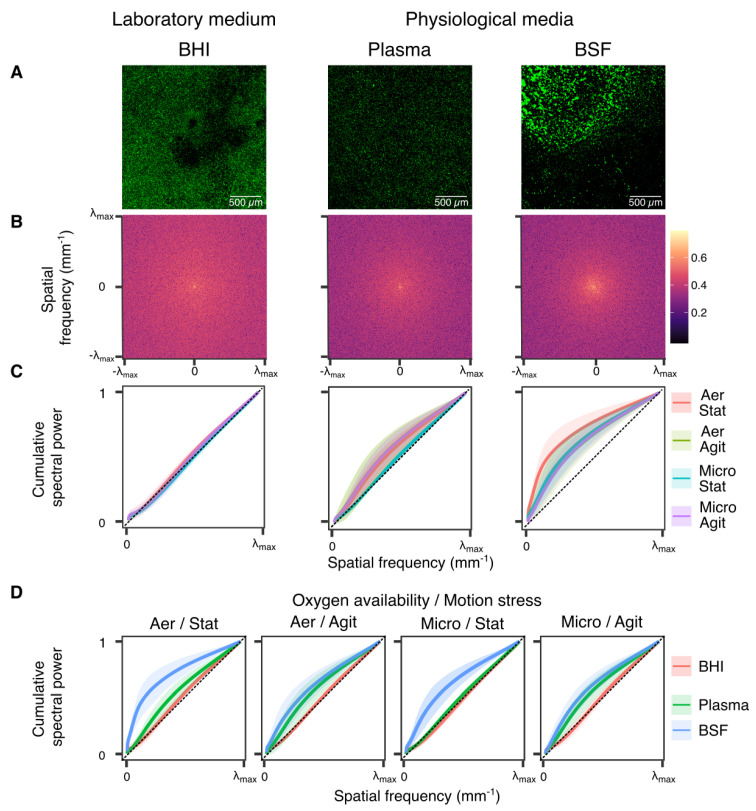
Fourier analysis reaffirms significant differences in biofilm pattern formation between lab and physiological media. (**A**) Representative CLSM images of sessile biofilm attachment to a titanium disk submerged in laboratory medium (BHI) or in physiological media (plasma, BSF). (**B**) Two-dimensional Fourier power spectrum of the representative CLSM images in (**A**). (**C**) CSPfs for the same medium in different oxygen availability/motion stress conditions. (**D**) CSPfs for the same oxygen availability/motion stress condition in different media.

**Figure 5 microorganisms-10-00526-f005:**
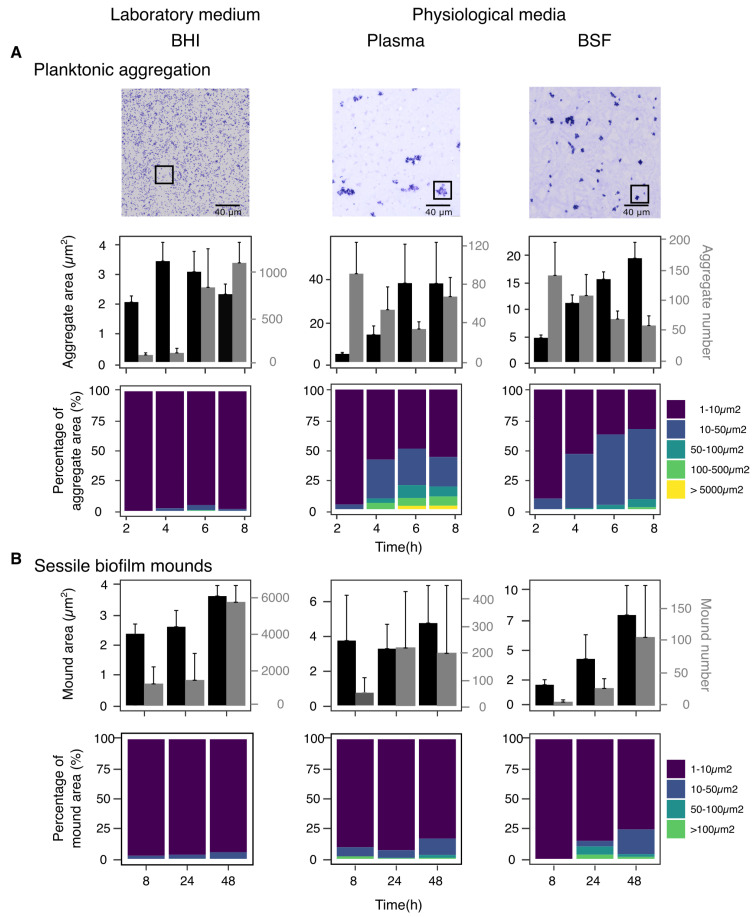
Kinetics of planktonic aggregation and sessile biofilm formation is highly sensitive to changes in growth medium. (**A**) Kinetic of planktonic aggregation in different media. Top panel: micrography of precipitated planktonic aggregates in different media. See Appendix A for a zoomed-in version of the image portions in the black squares, together with exemplative aggregate size measures. Mid panel: mean and standard deviation of aggregate size (black); mean and standard deviation of aggregate number (gray). Bottom panel: kinetic of percentage abundance of aggregates of different sizes. (**B**) Kinetic of sessile biofilm formation in different media. Top panel: mean and standard deviation of biofilm mound size (black); mean and standard deviation of biofilm mound number (gray). Bottom panel: kinetic of percentage abundance of biofilm mounds of different sizes.

**Figure 6 microorganisms-10-00526-f006:**
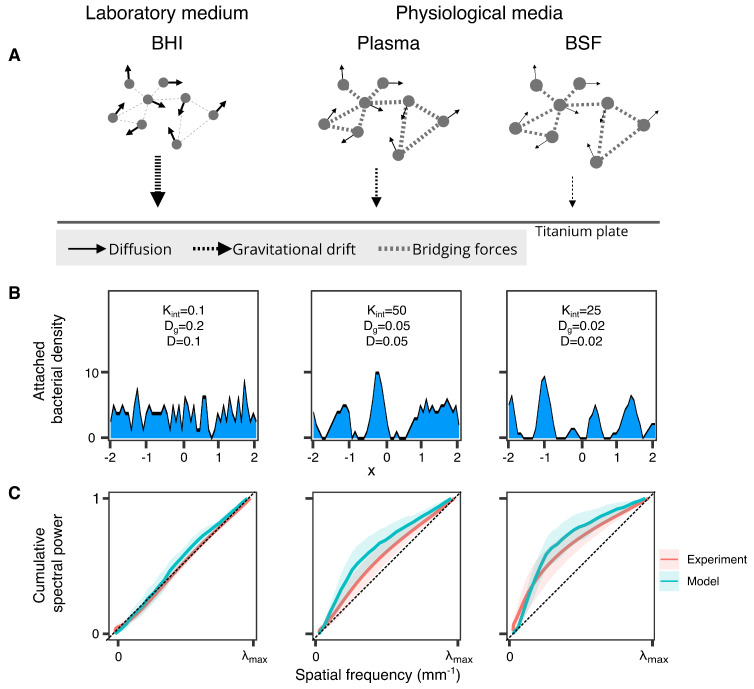
Modeling assumptions, model predictions, and comparison to experimental data. (**A**) Schematic model representation and differences between the simulated media. Thicker arrows or links mean larger physicochemical parameter. (**B**) Simulated sessile biofilm density. (**C**) Comparison between experimentally measured and simulated CSPfs. Thick line and shadow represent the mean and standard deviation of CSPf over replicas, respectively. Experimental CSPfs were averaged over oxygen availability and motion stress conditions.

**Table 1 microorganisms-10-00526-t001:** Physicochemical determinants of planktonic aggregation.

		Candidate Molecules [Concentration g/L] (Viscosity cP)	Ref.
Bridging forces	BHI	-	-
	Plasma	fibrinogen [2–4 g/L], albumin [30–45 g/L],	[21,22]
		fibronectin [0.2–0.4 g/L]	
	SF-H	fibrinogen [0–0.03 g/L], albumin [12 g/L], HA [2–4 g/L]	[23,24,25]
	SF-D	fibrinogen [0.2–0.6 g/L], albumin [-], HA [1–2 g/L]	[23,24,25]
Viscosity	BHI	-	-
	Plasma	fibrinogen, albumin—(1.1–1.3 cP)	[21]
	SF-H	fibrinogen, albumin, HA, proteoglycans—(40 cP)	[13]
	SF-D	fibrinogen, albumin, HA, proteoglycans—(6–40 cP)	[26]

## Data Availability

The data presented in this study are openly available in Zenodo at https://zenodo.org/record/5507271#.YW_OG55ByUk (accessed on 14 September 2021).

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
