# Peer review of "Host Environment Shapes S. aureus Social Behavior as Revealed by Microscopy Pattern Formation and Dynamic Aggregation Analysis"

_microorganisms, 2022, doi:10.3390/microorganisms10030526_

Round 1
Reviewer 1 Report
This is an interesting study on the influence of physiological conditions (i.e., growth medium, oxygen availability, motion stress) on the formation of planktonic aggregates and biofilm growth by S. aureus and their interaction. In general, the authors performed an adequate number of experiments to support their findings and the manuscript is well-written.
I only provide some minor comments, before this manuscript could be proceeded to publication.
L10, S. aureus should be written in italics
L203 & L208, Table ??
L284, Figure ??
L353, ica should be written in italics
L383, “37 oC”
L410, “overnight growth”
L412, “30 min”
Author Response
The authors thank the Reviewers for their comments and for the interest demonstrated in our research study.
The authors inform the Editor of Microorganisms that:
- Every changes in the revised document has been tracked in blue.
- The manuscript title has been modified according to the suggestions of the Reviewer#3
- 2 references have been added
- Figure 1 has been revised and a new Figure 1 will be submitted along with the modified manuscript
REVIEWER #1
This is an interesting study on the influence of physiological conditions (i.e., growth medium, oxygen availability, motion stress) on the formation of planktonic aggregates and biofilm growth by S. aureus and their interaction. In general, the authors performed an adequate number of experiments to support their findings and the manuscript is well-written.
The authors would like to thank Reviewer #1 for the positive comment and for the minor comments aimed at improving the quality of our manuscript.
I only provide some minor comments, before this manuscript could be proceeded to publication.
The authors thank the Reviewer#1 for all the minor corrections suggested.
L10, S. aureus should be written in italics
Modified as suggested
L203 & L208, Table ??
The authors apologize for the oversight. We inform the Reviewer#1 that the number of the Table has been added.
L284, Figure ??
We inform the Reviewer#1 that we corrected this line entering the Figure number.
L353, ica should be written in italics
Modified as suggested
L383, “37 oC”
The authors modified the Celsius symbol, as suggested.
L410, “overnight growth”
Modified as suggested
L412, “30 min”
Modified as suggested
Reviewer 2 Report
This paper describes a detailed analysis of sessile biofilm formed on a titanium surface compared to the formation of multicellular aggregates in the planktonic phase. The authors used optical and confocal microscopy combined with mathematical modelling to investigate the relationship between the planktonic and surface-bound phases and showed that they do not develop independently but are interlinked.
The paper extends the earlier observations by this group by focussing on a clinically relevant strain isolated from a prosthetic joint infection. The paper is well written, the data is sound and the paper makes a contribution to knowledge about the formation of biofilm under in vivo-like conditions bearing in mind that the bulk of studies on biofilm formed by S.aureus has been performed under artificial conditions in the laboratory (static growth, rich broth, no host proteins)
A few minor changes should be made
- Can the authors comment on the biological significance of their findings relating to the diversity of S.aureus strains that can cause PJ infection. Their study focusses on a single isolate with no information about its place in the overall phylogeny of the species
- Figure 1. Can the authors explain if the images refer to observations of the biofilm from above or if they represent a slice through the biofilm matrix
- In the discussion the authors might like to comment on the biological significance of published biofilm studies which demonstrated accumulation involving homophilic interactions between cell wall-anchored surface proteins. In some case PNAG is not involved
- English usage etc
- line 54. Staphylococcal
- line 62 biofilm
- line 76. Be more specific. lower rather than different ?
- line 117 led rather than lied
- line 152 evidence
- line 160 and legend to fig 5 kinetics
- line 208 insert table number
- line 247 evidence
- line 251 insert a between appear and long
- line 282. Led
- line 283 insert figure numbers
- line 293 involves
- line 300 insert an after as
- line 337 insert one after main
- line 344 insert a after as
- line 352 italicize ica
- line 360 insert at before the molecular level
Author Response
The authors thank the Reviewers for their comments and for the interest demonstrated in our research study.
The authors inform the Editor of Microorganisms that:
- Every changes in the revised document has been tracked in blue.
- The manuscript title has been modified according to the suggestions of the Reviewer#3
- 2 references have been added
- Figure 1 has been revised and a new Figure 1 will be submitted along with the modified manuscript
REVIEWER #2
This paper describes a detailed analysis of sessile biofilm formed on a titanium surface compared to the formation of multicellular aggregates in the planktonic phase. The authors used optical and confocal microscopy combined with mathematical modelling to investigate the relationship between the planktonic and surface-bound phases and showed that they do not develop independently but are interlinked.
The paper extends the earlier observations by this group by focusing on a clinically relevant strain isolated from a prosthetic joint infection. The paper is well written, the data is sound and the paper makes a contribution to knowledge about the formation of biofilm under in vivo-like conditions bearing in mind that the bulk of studies on biofilm formed by S. aureus has been performed under artificial conditions in the laboratory (static growth, rich broth, no host proteins)
We thank the Reviewer #2 for reading our research article and for all the suggestions that allowed us to further improve its quality (including the provided extensive proofreading).
A few minor changes should be made
Can the authors comment on the biological significance of their findings relating to the diversity of S. aureus strains that can cause PJ infection. Their study focusses on a single isolate with no information about its place in the overall phylogeny of the species
The authors would like to thank the Reviewer#2 for rising such an important issue. Indeed, S. aureus is known to exhibit a high genetic diversity. Despite this, past studies evidenced that all S. aureus strains are able to aggregate in presence of synovial fluids and to form large chunks of biofilm (the available bibliography is reported in the reference list). This might be explained by the impressive arsenal of surface binding proteins, often characterized by redundancy of function, as many surface expressed proteins were found able to bind fibrinogen and fibronectin (Crosby 2016, 10.1016/bs.aambs.2016.07.018). We recognize that both the lack of phylogenetic or genetic characterization and the employment of a single isolate represent a limitation of our study and for this reason this limitation is now stressed in the discussion (lines 264-269 and 348-352).
Despite this limitation, the aim of our work in the first place was to focus the attention on the fundamental issue of in vitro settings for biofilm studies to resemble ex vivo observations, and to contribute by developing a methodological procedure for the study and interpretation of such phenotypical changes. Based on these preliminary results, a further characterization of genotypically different strains would help define the molecular mechanisms at the basis of the observed phenotypes, possibly suggesting new therapeutic targets for the treatment of staphylococcal joint infections.
Figure 1. Can the authors explain if the images refer to observations of the biofilm from above or if they represent a slice through the biofilm matrix?
Images in Figure 1 show three-dimensional representations of staphylococcal biofilms as seen from above (lower panels, 5x) or inclined (upper panels, 20x) to highlight biomass differences of sessile biofilms in different media. This information is now integrated in the legend of Figure 1 as follows:
“Figure 1. Three-dimensional reconstruction of sessile biofilm formation on titanium surface by means of CLSM. (a) Inclined view and (b) top view of biofilm grown in the presence of laboratory medium BHI or in physiological fluids, plasma (left) and bovine synovial fluid (BSF, right). Upper panels show images acquired with a 20x objective; lower panels show images acquired with a 5x objective”
Please see the attachment.
In the discussion the authors might like to comment on the biological significance of published biofilm studies which demonstrated accumulation involving homophilic interactions between cell wall-anchored surface proteins. In some case PNAG is not involved.
We agree with the Reviewer#2 that PNAG’s role in biofilm formation and accumulation might have been overestimated by in vitro studies. We discussed this interesting aspect in the discussion (lines 292-294), by citing a recent study (Skovdal 2021, 10.1099/jmm.0.001287) which suggests that the assumed importance of polysaccharides for biofilm formation is an artifact of studying biofilms in laboratory media. In our previous study (Bidossi 2020, 10.3389/fmicb.2020.01368), we reported that plasmin was able to efficiently disrupt biofilm and aggregates formed by 6 different S. aureus and that proteinase K demonstrated an even more efficient action. This suggests that other mechanisms are involved in bacterial aggregation and biofilm formation, such as homophilic interactions of expressed surface proteins. S. aureus possesses many different cell wall-associated proteins which are known to have a role in cell–cell interactions during biofilm development, such as SraP, SdrC, SasG and even fibronectin binding proteins (Herman-Bausier 2015, 10.1128/mBio.00413-15; Speziale 2014, 10.3389/fcimb.2014.00171).
However, since the aim of the present work is to present an image pattern analysis supported by a mathematical predictive model, we did not investigated the staphylococcal behavior at genetic or proteomic level. This drawback is stressed in the revised version of the manuscript (lines 264-269 and 348-352) and, because of this limitation, we decided to see in the right perspective some assumptions referring to the cellular interactions mediating the bacterial aggregation process in the tested conditions. Hence, substantial modifications throughout the discussion section were made to contextualize the results obtained by this methodological analysis, highlighting the need to further investigate the biological pathways behind the described bacterial behavior. A reference to homophilic interactions of surface proteins was properly fitting in this section of the discussion and is now reported with other 2 references in lines 294-295.
English usage etc
The authors would like to thank the Reviewer#2 for reporting these grammatical and minor errors.
line 54. Staphylococcal
Modified as suggested
line 62 biofilm
Modified as suggested
line 76. Be more specific. lower rather than different ?
According to the Reviewer#2’s suggestion, we substitute “different” with a more specific word: “larger”.
line 117 led rather than lied
Modified as suggested
line 152 evidence
Modified as suggested
line 160 and legend to fig 5 kinetics
Modified as suggested
line 208 insert table number
Modified as suggested
line 247 evidence
Modified as suggested
line 251 insert a between appear and long
Modified as suggested
line 282. Led
Modified as suggested
line 283 insert figure numbers
Modified as suggested
line 293 involves
Modified as suggested
line 300 insert an after as
Modified as suggested
line 337 insert one after main
Modified as suggested
line 344 insert a after as
Modified as suggested
line 352 italicize ica
Modified as suggested
line 360 insert at before the molecular level
Modified as suggested

Reviewer 3 Report
In this manuscript, authors analyzed the bacterial growth characteristics in physiological microenvironment (physiological fluid), comparing with laboratory media, and revealed distinct differences. Although authors did extensive study, there are some basic concerns on this manuscript.
1.The objective of this study may be obvious and latest technology was employed, however, the study design seems to be primitive, and parameters of this study were too vast to draw scientific conclusion. That is, authors used media, BHI, plasma, BSF, which are mixtures of various different substances with different concentration. In such case, phenotypical difference is not able to be attributable to any specific factor(s),. Authors can describe just a difference of the "media". This appears to be very primitive study.
2. Description of bacterial strain is not clear. In methods section, "a clinically relevant strain isolated from an infected prosthetic knee..." is shown, but there is no strain name and no reference of the strain. Experiments using bacterial strain without name may not be believed as a scientifically sound research, probably by any journal. Authors wrote about this strain, as "multidrug resistant high biofilm producer S. aureus strain". Only in such explanation, it is really difficult to understand the characteristics of strain. Readers may be afraid that if other S. aureus strain is used, results of this study may be different. Therefore, authors could have used any established strain, of which genetic and phenotypic traits had been apparent.
3. Sentences are sometimes too long to understand the meaning. The manuscript should be substantially shortened and concise, using clear sentences.
Author Response
The authors thank the Reviewers for their comments and for the interest demonstrated in our research study.
The authors inform the Editor of Microorganisms that:
- Every changes in the revised document has been tracked in blue.
- The manuscript title has been modified according to the suggestions of the Reviewer#3
- 2 references have been added
- Figure 1 has been revised and a new Figure 1 will be submitted along with the modified manuscript
REVIEWER #3
In this manuscript, authors analyzed the bacterial growth characteristics in physiological microenvironment (physiological fluid), comparing with laboratory media, and revealed distinct differences. Although authors did extensive study, there are some basic concerns on this manuscript.
We thank the Reviewer #3 for reading our research article and for all the suggestions that allowed us to further improve its quality. Indeed, following the specific comments raised by the reviewer, we decided to extensively revise the whole article to highlight that the main aim of the study was to develop a methodology for biofilm image analysis, thus limiting the speculative conclusions.
1. The objective of this study may be obvious and latest technology was employed, however, the study design seems to be primitive, and parameters of this study were too vast to draw scientific conclusion. That is, authors used media, BHI, plasma, BSF, which are mixtures of various different substances with different concentration. In such case, phenotypical difference is not able to be attributable to any specific factor(s). Authors can describe just a difference of the "media". This appears to be very primitive study.
We agree with the Reviewer#3 that the experiments performed in the present study cannot allow us to draw definitive conclusions. Hence, we extensively revised the whole article in order to be less speculative. Now it is largely stated throughout the text that the main aim of the study was to improve the sensitivity of quantitative and qualitative biofilm images analysis. The obtained results allowed us to design a mathematical model that accurately predicts the biological observations. Despite that, there is the need to further validate the proposed model in future studies to explain the observed phenotypic differences in terms of specific factors. All the aforementioned limitations are now discussed in the revised discussion and contextualized within the aim of the manuscript which is methodological rather than descriptive.
2. Description of bacterial strain is not clear. In methods section, "a clinically relevant strain isolated from an infected prosthetic knee..." is shown, but there is no strain name and no reference of the strain. Experiments using bacterial strain without name may not be believed as a scientifically sound research, probably by any journal. Authors wrote about this strain, as "multidrug resistant high biofilm producer S. aureus strain". Only in such explanation, it is really difficult to understand the characteristics of strain. Readers may be afraid that if other S. aureus strain is used, results of this study may be different. Therefore, authors could have used any established strain, of which genetic and phenotypic traits had been apparent.
The authors would like to thank the Reviewer#3 for rising such an important issue. Indeed, S. aureus is known to exhibit a high genetic diversity. Despite this, past studies evidenced that all S. aureus strains are able to aggregate in presence of synovial fluids and to form large chunks of biofilm (the available bibliography is reported in the reference list). This might be explained by the impressive arsenal of surface binding proteins, often characterized by redundancy of function, as many surface-expressed proteins were found able to bind fibrinogen and fibronectin (Crosby 2016, 10.1016/bs.aambs.2016.07.018). We recognize that both the lack of phylogenetic or genetic characterization and the employment of a single isolate represent a strong limitation of our study and this is now clearly stated in discussion (lines 264-269 and 348-352).
Despite this limitation, the aim of our work in the first place was to focus the attention on the fundamental issue of in vitro settings for biofilm studies to resemble ex vivo observations, and to contribute by developing a method for the study and interpretation of such phenotypical changes.
3. Sentences are sometimes too long to understand the meaning. The manuscript should be substantially shortened and concise, using clear sentences.
The authors would like to thank Reviewer#3 for this comment that will certainly help to improve quality of our manuscript. We proofread the article and we hope that the modifications we made were sufficient to express the concepts in a clearer way.
Round 2
Reviewer 3 Report
The revised version has been well improved.